# Neural Mechanisms of the Conscious and Subliminal Processing of Facial Attractiveness

**DOI:** 10.3390/brainsci13060855

**Published:** 2023-05-25

**Authors:** Xuejiao Hou, Junchen Shang, Shuo Tong

**Affiliations:** 1College of Education, Suihua University, Suihua 152061, China; houxuejiao1224@163.com; 2Department of Medical Humanities, School of Humanities, Southeast University, Nanjing 211189, China; 3College of Psychology, Liaoning Normal University, Dalian 116029, China; 4Beijing Key Laboratory of Learning and Cognition, School of Psychology, Capital Normal University, Beijing 100048, China; 2203502066@cnu.edu.cn

**Keywords:** facial attractiveness, conscious, subliminal, backward masking, event-related potentials

## Abstract

The purpose of this study was to investigate the neural activity evoked by facial attractiveness in conscious and subliminal conditions. The 41 participants judged facial attractiveness in a conscious condition and a subliminal condition (backward masking paradigm). The event-related potential (ERP) analysis indicated that in the conscious condition, more attractive faces elicited a longer N1 (80–120 ms) latency than less attractive faces. Moreover, more attractive female faces elicited a larger late positive component (LPC) (350–550 ms) amplitude than less attractive female faces. In the subliminal condition, more attractive faces elicited a longer P1 (140–180 ms) latency than less attractive faces. The present study demonstrated that more attractive faces evoked different early-stage ERPs from that evoked by less attractive faces in both conscious and subliminal conditions. However, the processing of facial attractiveness is obviously weakened in the subliminal condition.

## 1. Introduction

“Is this person beautiful?” This question often comes to mind when we inquire about or meet a stranger. Facial beauty (often equivalent to facial attractiveness in most research) is an important contributor to the overall rating of attractiveness [1]. Specifically, there is a positive correlation between attractiveness ratings and facial features, such as large eyes, small nose, small chin, prominent cheekbones, and narrow cheeks [2]. People often have a preference for someone with an attractive face. For example, individuals with attractive faces have been considered to have more positive personality traits [3], and they tend to obtain higher salaries [4]. From an evolutionary perspective, facial attractiveness judgment is correlated with health and mate selection [5]. Facial attractiveness is determined by biologically evolutionarily important characteristics, such as symmetry, average and above average sexual dimorphic characteristics [5,6,7], skin condition [8], and age [7], although there was a debate about averageness suggesting that highly attractive facial shapes were not average [9]. Although facial attractiveness has been extensively studied in conscious perception, it is still not clear whether facial attractiveness can be processed below the threshold of awareness.

Many researchers have used functional magnetic resonance imaging (fMRI), which is used to test neural activity when a subject is doing a particular task by measuring hemodynamic signal changes with high spatial resolution, to explore the cognitive processing of facial attractiveness. Although the neural activities elicited by attractiveness varied across different experimental tasks, most studies have suggested that attractive faces induced different activation compared with unattractive faces in the reward regions of the brain [10,11,12], including the amygdala [13], the anterior cingulate cortex [11,14,15,16], the nucleus accumbens [14,15,17], the orbitofrontal cortex [10,11,12,14,15,16,18], the caudate nucleus [10,15], the medial prefrontal cortex [14,18], the putamen [12,15], and the ventral tegmental area [15].

Nevertheless, the temporal resolution of fMRI is low, and many researchers have used another technology with higher temporal resolution but low spatial resolution to investigate the temporal dynamics of the processing of facial attractiveness. Event-related potentials (ERPs) measure neurophysiological signals and can show various periods of cognitive processing. The earliest ERP component elicited by facial attractiveness is N1, which is related to the perceptual processing of attractiveness features and facial structure detection [19]. Among studies using the attractiveness judgment task, Schacht et al. found that attractive faces and unattractive faces induced larger early components at approximately 150 ms and a larger late positive component (LPC) than neutral faces [20]. However, Marzi and Viggiano found that highly attractive faces elicited larger N170 and P1 than less attractive faces [21]. In addition, when participants were asked to detect a face among many objects and then judge the facial attractiveness, attractive faces elicited larger P1 than unattractive faces [22]. Studies using tasks unrelated to attractiveness judgment also found that facial attractiveness resulted in differences in the P1 [21,23], N170 [21], and LPC [24,25,26,27,28]. Moreover, some decision-making studies found that unattractive faces elicited a larger N2 than attractive faces [25,28,29], although Chen et al. suggested that attractive faces induced a larger N2 than unattractive faces [24]. Thus, the processing of facial attractiveness elicited early components and late components, despite inconsistent differences in ERPs elicited by faces varying in attractiveness. Previous studies also analyzed the latency of ERP components elicited by facial attractiveness, but the results were controversial. Peak latency is a reflection of the duration from stimulus onset and represents the processing speed [22]. The latencies of N1, P2, N2, or N3 elicited by attractive and unattractive faces were not different in an attractiveness judgment task, a recognition task, or a decision-making task [23,24]. Zhang and Deng also did not show differences in the latencies of N170, P2, and N3 elicited by attractive and unattractive faces but found that in male participants, attractive faces elicited a longer P3b latency than unattractive faces [22].

Most studies have investigated the neural mechanisms of the processing of attractiveness when faces are fully visible. Some behavioral studies have explored the subliminal processing of facial attractiveness using the masking paradigm, one of the methods used for subliminal presentation [30,31,32,33]. Masking can eliminate the visibility of a face by briefly presenting and showing geometrical shapes (“masks”) that precede and follow the face. For example, using forward and backward masking paradigms, Olson and Marshuetz found that ratings for attractive faces were higher than those for unattractive faces when the faces were presented for 13 ms, suggesting that the preference for attractive faces can occur automatically and without apparent awareness [34]. In contrast, Tsikandilakis, Bali and Chapman. did not observe subliminal processing of facial attractiveness, and the evaluation of facial attractiveness required conscious awareness when the face was presented for 33.33 ms in a backward masking condition [35]. It is not known, in particular, whether what was demonstrated in the conscious processing of facial attractiveness can also be demonstrated in subliminal processing at the neural level.

The objective of this study was to examine this question. Although much progress has been made in research on the neural processing of facial attractiveness, it remains unclear whether a similar pattern of neural activity induced by facial attractiveness is maintained in the subliminal condition compared with the conscious condition. Unconscious processes have a potential influence on higher mental processes [30]. Thus, subliminal processing of facial attractiveness may be the basis of preference for facial attractiveness in conscious conditions. It is important to investigate how the subliminal processing of facial attractiveness works since it can help explain the neural mechanism of the preattentive appraisal of facial beauty. Moreover, per the inconsistencies of studies using the masking paradigm [34,35], it is necessary to explore the neural activities underlying the processing of facial attractiveness in subliminal conditions.

In addition, one critical and possible confound that was not addressed in previous research about facial attractiveness was emotion. Prior research mainly rated the attractiveness of faces only. Although some researchers rated emotion (i.e., pleasure, arousal) in addition to attractiveness and reported that the ratings of pleasure of attractive faces were greater than those of unattractive faces [23], they did not rule out the confounding effects of emotion. The results of previous studies on facial attractiveness can be explained under the framework of emotion processing as well; that is, attractive faces elicited positive emotions, and unattractive faces elicited negative emotions. Moreover, emotional information has been shown to allow subliminal processing in the masking paradigm [36]. In a previous study, the target face was presented for approximately 16.67 ms in a backward masking task, and the mask was subsequently presented for 150 ms. The results showed that happy faces elicited smaller P1, while sad faces elicited larger P1, compared to neutral faces for patients with depression [37]. Peng, Cui, Wang and Jiao used the same procedure and found that for individuals with internet gaming disorder, happy faces elicited larger N170 than neutral faces [38]. We tried to rule out the possible influence of emotion to determine whether processing of facial attractiveness indeed existed in the masking task. However, there are not truly any faces for which each attribute is balanced. We only balanced the pleasure and arousal of more attractive faces and less attractive faces at the expense of small differences in attractiveness ratings. The dominance of faces and other attributes was not controlled. Indeed, it is difficult to experimentally unconfound variations of faces on all the other social dimensions [39].

Adopting an attractiveness judgment task in the conscious condition and a backward masking paradigm in the subliminal condition, the present study used ERPs to investigate the differences in neural activity induced by more attractive faces and less attractive faces. In masking paradigms, the shorter the time that the target stimulus is presented, the more likely it is to be invisible to participants. To date, the shortest presentation time for faces is 10 ms in backward masking [40]. Therefore, in the present study, the faces were presented for 10 ms. The participants were asked to evaluate the attractiveness of faces in a conscious condition and a subliminal condition. N1, N2, and P1 components were analyzed to compare the temporal dynamics of brain activity evoked by facial attractiveness in the early stage, while the N3 and LPC amplitudes were analyzed in the late stage. We hypothesized that more attractive faces would induce different neural activities compared with less attractive faces both in the conscious condition and in the subliminal condition, while the differences in neural activities would be attenuated in the subliminal condition.

## 2. Materials and Methods

### 2.1. Participants

The present study used G*Power 3.1.9.4 to estimate the sample size [41]. At least 24 participants were required to obtain a medium effect size (*f* = 0.25) with 80% power at a significance level of 0.05. To prevent possible loss of participants, forty-one college students (21 females, *M_age_* = 21.56, *SD_age_* = 2.47) were recruited. All participants were physically and mentally healthy and were all right-handed, with normal or corrected-to-normal vision. None of the individuals had participated in any similar experiments before. All participants signed informed consent before the experiment. The experiment was conducted in accordance with the Declaration of Helsinki and was approved by the Ethics Committee of Liaoning Normal University. Six participants were excluded from the analysis for the following reasons. The accuracy of four participants in consciousness detection was outside the range of the binomial distribution, suggesting that the perception of the faces was not subliminal. The valid ERP data for one participant were not sufficient for averaging. One participant pressed the wrong key, which caused a program error. The main analysis consisted of 35 participants (17 females).

### 2.2. Materials

A total of 125 male faces and 119 female faces were taken from the Baidu website in 2016. The search keywords were images (照片/证件照 in Chinese). We chose pictures in which the people in the images had a neutral emotional expression, were visible from the frontal view and had a forward eye gaze, and were not wearing glasses or earrings. The images had previously been cropped and presented on a gray background using Photoshop 8.0.1. The size of all images was approximately 4.2° × 6.7°. Standardized facial stimuli were developed and validated within Wang, Tong, Shang and Chen’s study [42], from which we selected faces based on pleasure, arousal, and attractiveness ratings. Their rating task was as follows. Thirty-three participants (17 females) who did not participate in the ERP experiment were asked to rate the pleasure and arousal, based on their gut feelings about the face, on the 9-point Self-Assessment Manikin (SAM) (1 = most unpleasant/the least awake or the least excited, 9 = most pleasant/the most awake or the most excited) [43]. In addition, the participants were asked to evaluate facial attractiveness on a 7-point scale (1 = not attractive at all, 4 = neutral, 7 = extremely attractive). The participants were asked after the rating task if they had seen these pictures before. All participants reported that they were unfamiliar with the faces and had not seen the faces before. Similar to Zhang et al. [23], we found that faces with higher pleasure and arousal scores also had higher attractiveness scores, whereas faces with lower pleasure and arousal scores had lower attractiveness scores. To obtain two groups of faces that differed significantly in attractiveness ratings but did not differ in pleasure and arousal, we selected faces with lower attractiveness ratings from the more attractive faces and the faces with higher attractiveness ratings from the less attractive faces, resulting in the small gap between the ratings of more attractive face stimuli and less attractive face stimuli. Thirty more attractive faces and thirty less attractive faces (half male and half female) were used as stimuli in the ERP experiment. Three separate one-way ANOVAs were performed on the arousal, pleasure, and attractiveness ratings. There was a significant effect of attractiveness (*F*(1, 58) = 68.76, *p* < 0.001, η_p_^2^ = 0.542, and 95% CI [0.41, 0.67]). There was no significant difference in arousal (*F*(1, 58) = 1.62, *p* = 0.208, η_p_^2^ = 0.027, and 95% CI [−0.08, 0.35]), or pleasure (*F*(1, 58) = 3.56, *p* = 0.064, η_p_^2^ = 0.058, and 95% CI [−0.01, 0.19]). The descriptive values are shown in Table 1.

The visual angle of the face stimuli was approximately 3.5° × 4.0°. Each face was rotated to create 60 inverted faces. Using MATLAB R2012b, each face was divided into small blocks of 8 × 8 pixels, and then the order of the small blocks was shuffled and randomly combined to generate 60 scrambled faces as masking stimuli. In addition, 10 pictures were selected as practice stimuli in the conscious condition, and 10 pictures were selected as practice stimuli in the subliminal condition, which were not shown in the ERP experiment. The descriptive values of practice stimuli are shown in Table 2.

The experimental paradigm was programmed using E-Prime 2.0. The stimuli were presented on an AOC LCD monitor (1024 × 768 pixels) with a width of 59.8 cm and a height of 33.6 cm. The refresh rate was set at 100 Hz. The experiment was carried out in a dark and quiet room. The participants were comfortably seated with their eyes at approximately 57 cm from the screen, and their heads were fixed on a chin rest.

### 2.3. Design

A 2 (facial attractiveness: more attractive and less attractive) × 2 (sex of the face: male and female) within-subjects design was employed. The peak amplitudes and the peak latencies of the N1, N2, and P1 components and average amplitudes of the N3 and LPC components were dependent variables.

### 2.4. Procedure

The experiment consisted of two parts, judging facial attractiveness in the conscious condition and subliminal condition. The order of the tasks was counterbalanced across the participants.

The conscious rating task was adapted from Zhang et al. [23] (see Figure 1A). In each trial, a fixation cross (0.08° × 0.06°) appeared in the center of the screen for 500 ms, followed by the presentation of a blank screen for 400–600 ms, then a face was presented for 1000 ms, followed by a blank screen for 1500 ms. The participants were asked to judge whether each face was “more attractive” or “less attractive” as quickly and accurately as possible by pressing “A” or “L” on the keyboard, respectively. A blank screen was shown after participants pressed the button; then, the next trial started. There were 60 faces. Each face was repeated four times. The experiment consisted of 240 trials (4 blocks of 60 trials each), and each face was presented once in each block. The faces were presented in randomized order. The participants rested for a few minutes between blocks. There were 10 practice trials before the formal experiment.

The subliminal rating task was adapted from Peng et al. [38] and Almeida et al. [40] (see Figure 1B). At the beginning of each trial, a fixation cross (0.08° × 0.06°) was presented for 500 ms, followed by the presentation of a blank screen for 400–600 ms. Then, a face was presented for 10 ms (it is notable that we used a monitor set at 100 Hz. We neither ensured whether it was actually a 100 Hz monitor nor checked for dropped frames. https://groups.google.com/g/psychopy-users/c/Xn7U6E_8h9g?pli=1 (accessed on 26 February 2013)), followed immediately by a 150 ms presentation of a scrambled face as a mask. On the basis of their gut feelings, the participants were asked to guess whether each face was “more attractive” or “less attractive” as quickly and accurately as possible by pressing “A” or “L”, respectively. Then, the participants were asked to answer whether they saw a face by pressing “A” or “L”. There were 60 faces in the experiment, and each face was presented 4 times, resulting in 240 trials divided into 4 blocks. Each face was presented once in each block. The faces were presented in randomized order. There were 20 practice trials before the formal experiment. The participants rested for a few minutes between blocks. After the subliminal task, a face awareness task (see Figure 1C) was used to measure the extent to which participants were aware of the faces. The procedure was similar to the subliminal rating task, except that participants were asked to guess the orientation of the face. The test consisted of 120 trials (60 upright faces and 60 inverted faces). Each face was presented once in an upright manner and once in an inverted manner. The faces were presented in randomized order. The electroencephalogram (EEG) was not recorded in this task.

### 2.5. ERP Recording and Analysis

All required information for ERP analyses is reported according to Keil et al. [44]. EEG signals were recorded (bandpass 0.01–30 Hz, sampling rate 500 Hz) through a 64-electrode scalp cap using the 10–20 system (Brain Products, Gilching, Germany). The left and right mastoids were used as the reference. The AFz channel was used as the ground electrode. An electrode was used to measure the electrooculogram (EOG) data. Impedance was kept below 10 kΩ. The averaging of ERPs was computed offline. EEG data were preprocessed by using Analyzer 2.0. Data were re-referenced offline to the mastoids TP9 and TP10 electrodes. Independent component analysis was applied to correct artifacts of blink. The ERPs induced by face stimuli were analyzed in the conscious and subliminal conditions. The epochs consisted of 200 ms before and 1000 ms after the onset of the face stimuli. ERPs were aligned to a 200 ms baseline. The epochs contaminated by eye blinks or muscle potentials exceeding ± 80 μV were excluded from averaging.

Based on previous studies [21,23,24,25,45] and waveforms in the present study, 15 electrodes were selected for analysis: F3, Fz, and F4 (frontal sites); FC3, FCz, and FC4 (frontal-central sites); C3, Cz, and C4 (central sites); CP3, CPz, and CP4 (central-parietal sites); and P3, Pz, and P4 (parietal sites). The following components and time windows for the final analysis were chosen: N1 (80–120 ms), P1 (140–180 ms), N2 (180–250 ms), N3 (260–300 ms), and LPC (350–550 ms). Figure 2 illustrates the electrodes analyzed. 2 (attractiveness: more attractive and less attractive) × 2 (sex of the face: male and female) × 3 (laterality: left sites, midline sites, and right sites) × 5 (location: Frontal, frontal-central, central, central-parietal and parietal) repeated-measures ANOVAs were conducted for the conscious and subliminal conditions. The dependent variables were the peak amplitudes and peak latencies of N1, P1, and N2 and the average amplitudes of N3 and LPC. Greenhouse–Geisser corrections were applied where needed.

## 3. Results

### 3.1. Face Awareness

The percentage of correct performance in the face awareness task was one criterion used for selecting participants for the main analysis. A binomial distribution was used to determine the criteria [46]. According to the formula of a binomial distribution, μ = np = 60, σ = $\sqrt{\mathrm{npq}}$ = 5.48, the upper limit of accuracy was 0.5751 ((μ + 1.645σ)/120), and the lower limit was 0.4249 ((μ − 1.645σ)/120). Four participants were removed from the analysis because they correctly guessed the orientation in more than 57.51% of the trials or correctly guessed the orientation in less than 42.49% of the trials (better or worse than chance, *p* < 0.05 for each tail). This criterion was used to ensure, as strictly as possible, that the faces were not consciously perceived.

Moreover, in the subliminal rating task, the trials in which the participants reported that they saw the face were also excluded. A total of 269 trials were excluded (accounting for 3.2% of the total trials).

### 3.2. ERP Results

The ERP wave map is shown in Figure 3. The ERP topographic map is shown in Figure 4.

#### 3.2.1. N1

Conscious condition:

The peak amplitudes of N1: There was no significant main effect of attractiveness or any interaction effects related to attractiveness (*F*s < 2.00, *p*s > 0.16).

The peak latencies of N1: The main effect of attractiveness was significant (*F*(1, 34) = 5.91, *p* = 0.021, η_p_^2^ = 0.148). More attractive faces (*M* = 104.41, *SE* = 1.31) elicited a longer N1 latency than less attractive faces (*M* = 102.19, *SE* = 1.48). The interactions related to attractiveness were not significant (*F*s < 2.29, *p*s > 0.10).

Subliminal condition:

The peak amplitudes of N1: There was no significant main effect of attractiveness or any interaction effects related to attractiveness (*F*s < 1.51, *p*s > 0.23).

The peak latencies of N1: There was no significant main effect of attractiveness or any interaction effects related to attractiveness (*F*s < 1.84, *p*s > 0.17).

#### 3.2.2. P1

Conscious condition:

The peak amplitudes of P1: There was no significant main effect of attractiveness or any interaction effects related to attractiveness (*F*s < 1.99, *p*s > 0.16).

The peak latencies of P1: There was no significant main effect of attractiveness or any interaction effects related to attractiveness (*F*s < 2.87, *p*s > 0.05).

Subliminal condition:

The peak amplitudes of P1: There was a significant attractiveness by sex of the face by laterality interaction effect (*F*(2, 68) = 3.17, *p* = 0.048, η_p_^2^ = 0.085). Paired-sample t-tests showed that in each laterality, there were no significant differences in the peak amplitude elicited by faces with different attractiveness ratings for each sex (*t*s < 1.40, *p*s > 0.17). The main effect of attractiveness and the interactions between attractiveness and the other factors were not significant (*F*s < 1.79, *p*s > 0.14).

The peak latencies of P1: There was a significant attractiveness by location interaction effect (*F*(2.59, 88.18) = 3.25, *p* = 0.032, η_p_^2^ = 0.087). A paired-sample t-test showed that more attractive faces elicited a longer P1 latency than less attractive faces at central-parietal sites (*t*(34) = 2.08, *p* = 0.046, Cohen’s *d* = 0.35), but differences were not significant in other brain regions (*t*s < 1.41, *p*s > 0.17). There was also a significant attractiveness by sex of the face by laterality interaction effect (*F*(2, 68) = 3.61, *p* = 0.032, η_p_^2^ = 0.096). Paired-sample t-tests showed that more attractive female faces elicited a longer P1 latency than less attractive female faces at the left sites (*t*(34) = 2.09, *p* = 0.044, Cohen’s *d* = 0.35). In addition, in the right and midline sites, there were no significant differences in the latency elicited by faces with different attractiveness levels for each sex (*t*s < 1.20, *p*s > 0.24). The main effect of attractiveness and the interactions between attractiveness and the other factors were not significant (*F*s < 2.74, *p*s > 0.07).

#### 3.2.3. N2

Conscious condition:

The peak amplitudes of N2: There was a significant attractiveness by sex of the face by laterality by location interaction effect (*F*(3.61, 122.57) = 2.64, *p* = 0.042, and η_p_^2^ = 0.072). Subsequent simple effect analysis was conducted, and for male faces, a 2 (attractiveness: more attractive and less attractive) × 3 (laterality: left, midline, and right sites) × 5 (location: Frontal, frontal-central, central, central-parietal and parietal) repeated-measures ANOVA showed that the main effect of attractiveness and the interactions between attractiveness and other factors were not significant (*F*s < 1.76, *p*s > 0.18). For female faces, the same repeated-measures ANOVA showed that the main effect of attractiveness and the interactions between attractiveness and the other factors were also not significant (*F*s < 1.14, *p*s > 0.33).

The peak latencies of N2: There was a significant attractiveness by sex of the face by laterality interaction effect (*F*(2, 68) = 7.48, *p* = 0.001, η_p_^2^ = 0.18). Paired-sample t-tests showed that there was no significant difference between more attractive faces and less attractive faces for each sex in each laterality (*t*s < 1.30, *p*s > 0.20). The main effect of attractiveness and the interactions between attractiveness and the other factors were not significant (*F*s < 3.35, *p*s > 0.07).

Subliminal condition:

The peak amplitudes of N2: There was no significant main effect of attractiveness or any interaction effects related to attractiveness (*F*s < 1.76, *p*s > 0.18).

The peak latencies of N2: There was no significant main effect of attractiveness or any interaction effects related to attractiveness (*F*s < 1.51, *p*s > 0.22).

#### 3.2.4. The Average Amplitude of N3

Conscious condition:

There was no significant main effect of attractiveness or any interaction effects related to attractiveness (*F*s < 2.97, *p*s > 0.09).

Subliminal condition:

There was no significant main effect of attractiveness or any interaction effects related to attractiveness (*F*s < 1.26, *p*s > 0.27).

#### 3.2.5. The Average Amplitude of LPC

Conscious condition:

There was a significant attractiveness by sex of the face interaction effect (*F*(1, 34) = 7.96, *p* = 0.008, η_p_^2^ = 0.19). Paired-sample t-tests showed that more attractive female faces elicited a larger LPC amplitude than less attractive female faces (*t*(34) = 2.24, *p* = 0.032, Cohen’s *d* = 0.38). There was no significant difference between the amplitude of LPC elicited by more attractive and less attractive male faces (*t*(34) = 0.91, *p* = 0.372, Cohen’s *d* = 0.15). In addition, the interaction effect between attractiveness and laterality was significant (*F*(2, 68) = 5.17, *p* = 0.008, η_p_^2^ = 0.132). Paired-sample t-tests showed that there were no significant differences between the LPC amplitude elicited by more attractive and less attractive faces at the left, midline, or right sites (*t*s < 1.41 *p*s > 0.16). There was no significant main effect for attractiveness or any other interaction effects related to attractiveness (*F*s < 1.46, *p*s > 0.23).

Subliminal condition:

There was no significant main effect of attractiveness or any interaction effects related to attractiveness (*F*s < 2.07, *p*s > 0.15).

## 4. Discussion

In the present study, we compared the neural activities of participants when they made judgments of facial attractiveness in a conscious condition and a subliminal condition.

In the conscious condition, more attractive faces elicited a longer N1 latency than less attractive faces. N1 is related to the perceptual processing of attractiveness features and facial structure detection [19]. Of note, peak latency represents the processing speed [22]. Our result was consistent with that of Zhang et al., who found that participants took more time to respond to attractive faces than to unattractive faces. Zhang et al. suggested that attractive faces captured more attention [23]. Moreover, participants were more cautious in judging attractive faces and thus took more time to respond to attractive faces. However, our result was inconsistent with Marzi and Viggiano, who reported that the participants responded faster to extremely attractive and extremely unattractive faces than to faces of medium and relatively low attractiveness [21]. The reason may be that the faces were presented for 1000 ms in the present study, and the participants had enough time to distinguish the level of facial attractiveness. However, in Marzi and Viggiano’s study, faces were presented for 500 ms, which is relatively short, and participants were able to differentiate easily distinguishable faces, such as extremely attractive and extremely unattractive faces.

In addition, the trend of our latency result was partially consistent with that of Zhang and Deng. They reported that attractive faces elicited a longer P3b latency (which is specific to their oddball task) than unattractive faces, but only in male participants [22]. However, the differences in N1 latency were inconsistent with some studies [23,24,47] that did not report differences in latency elicited by differing attractiveness levels. This may be because Lu et al. used cartoon faces [47], which have lower ecological validity than actual faces. The pleasure of the face stimuli was not controlled for in Zhang et al.’s [23], which may have led to these inconsistencies. Instead of using an attractiveness judgment task, Chen et al. used a decision-making task [24]. The participants were focused on the investment decision when faces were presented, which limited the cognitive resources available for the faces. Thus, there was no difference in latency elicited by attractiveness.

In the conscious condition, we also found that more attractive female faces elicited a larger LPC than less attractive female faces. This result was consistent with previous findings [23,25,27,28], which revealed that attractive faces elicited a larger LPC than unattractive faces. Attractive faces draw more attention, thus evoking a larger LPC [28]. The LPC is also thought to reflect different cognitive functions [23]. In addition, greater LPC induced by more attractive female faces also suggested that these faces were easier to process in the later stage because they are more fluent or congruent with our prototypical expectation [48]. The difference in the LPC was found only in female faces; this may be because females with more attractive faces have good genes and a high probability of successful reproduction [23]. This is also in line with an eye movement study that found that the differences in fixation time between attractive and unattractive female faces were greater than the differences between attractive and unattractive male faces [49]. However, our result was inconsistent with some studies [20,24,50]. Schacht et al. found that attractive faces and unattractive faces elicited a larger LPC than faces with medium attractiveness [20]. However, the facial expressions were not controlled, and the LPC was possibly elicited by the facial expressions. The present study showed that more attractive female faces still elicited larger LPC even when pleasure and arousal were controlled for. Therefore, the difference in LPC may simply indicate the difference in the processing of attractiveness and not the emotion of the face. Chen et al. found that unattractive faces elicited a larger LPC than attractive faces [24]. A possible reason for this discrepancy was that different tasks were used in the present study (attractiveness judgment task) and Chen et al. (decision-making task). Muñoz and Martín-Loeches found that both attractive and unattractive stimuli elicited a larger LPC than neutral stimuli [50]. However, there were both face and body pictures in their study, and facial attractiveness was not separately analyzed, which might have caused the inconsistency in the LPC data.

However, no difference was found in the latency of N1 in the subliminal condition. Instead, more attractive faces elicited a longer P1 latency than less attractive faces at central-parietal sites, which occurred later than in the conscious condition, most likely due to the delay in the processing caused by masking. The P1 component may reflect the processing of facial structures and features, which has been related to attention mechanisms [51]. The longer P1 latency elicited by more attractive faces may have been because the more attractive faces captured the attention of the participants and prolonged the reaction time. Another possible reason was that we strictly controlled the pleasure and arousal of the faces, which reduced the difference in attractiveness between more attractive faces and less unattractive faces. When the faces were presented for only 10 ms and masked, it was more difficult for the participants to judge the attractiveness of the faces. Therefore, the participants responded slower to more attractive faces. Nonetheless, the results suggested that even if the participants were not aware of the face, facial attractiveness could still be rapidly processed in the early stage, which was reflected in the P1 latency. In addition, more attractive female faces elicited a longer P1 latency than less attractive female faces at the left sites, suggesting that the participants were more sensitive to the attractiveness of female faces.

In summary, in the conscious condition, facial attractiveness elicited differences in both early and late ERP components, while in the subliminal condition, facial attractiveness elicited differences only in early ERP components. The current study provides ERP evidence for previous studies that facial attractiveness can be perceived in subliminal conditions [34]. Moreover, there were only differences in ERP latencies in the subliminal condition rather than differences in amplitudes. These data showed that the processing of facial attractiveness was weakened in the subliminal condition, in which the faces were presented for an extremely short time and were invisible, which may have resulted in the weakened processing of facial attractiveness. In addition, there was a difference in the N1 latencies in the conscious condition, whereas, in the subliminal condition, facial attractiveness only elicited a difference in P1 latencies. The effect of attractiveness appeared later in the subliminal condition, which may have indicated that masking stimuli impeded the processing of facial attractiveness.

There were some limitations in the current study. Firstly, in the face rating task, the SAM scale was used to capture participants’ emotional reactions. The SAM scale consists of three dimensions: the values of pleasure, arousal, and dominance. However, in this experiment, we adopted ratings from Wang et al.’s [42], in which only pleasure and arousal were used. Dominance may have affected the results of the experiment. Secondly, the gap in attractiveness of our face stimuli was small since we controlled the pleasure dimension of face stimuli. Actually, a neutral expression did not mean that the pleasure was neutral. Future research could ask participants to judge facial expressions (1-positive, 2-neutral, 3-negative) rather than pleasure and select neutral faces to increase the sample of stimuli. Thirdly, in the subliminal condition, a face awareness task (one of the forced-choice tasks) was used to exclude the participants who participants were aware of the faces. We also adopted the binomial distribution formula to calculate consciousness detection accuracy according to Anderson et al.’s [46]. However, according to the signal detection theory, forced-choice tasks are only suitable for measuring sensitivity rather than response bias [52]. Therefore, the accuracy of the face awareness task cannot strictly differentiate between subliminal perception and conscious perception. Future research should apply signal detection theory to measure awareness of the stimuli. Another problem of our method is that participants’ accuracy at the chance level in the face awareness task did not mean their performance was indeed at a subliminal level [53]. Dienes suggested a more reasonable approach to ask other group participants to identify stimuli that are difficult to observe but still provide some level of conscious experience. Then, we can know the threshold of conscious detection. Lastly, during the experiment, we set the refresh rate of the monitor to 100 Hz. However, we did not check for dropped frames. It is possible that there were dropped frames and the target faces were presented for longer than 10 ms. Future research should use a more accurate method to control for dropped frames.

## 5. Conclusions

More attractive faces evoked different early-stage ERPs from that evoked by less attractive faces in both conscious and subliminal conditions. However, the processing of facial attractiveness is obviously weakened in the subliminal condition.

## Figures and Tables

**Figure 1 brainsci-13-00855-f001:**
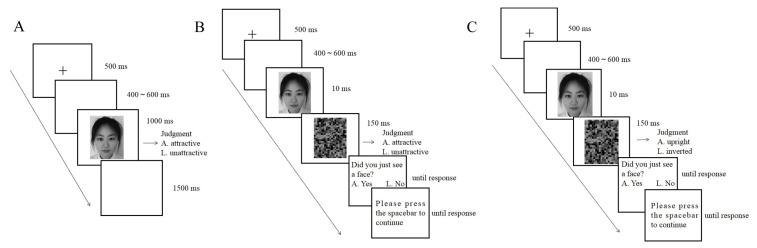
(**A**) The procedure of the conscious rating task. (**B**) The procedure of the subliminal rating task. (**C**) The procedure of the face awareness task.

**Figure 2 brainsci-13-00855-f002:**
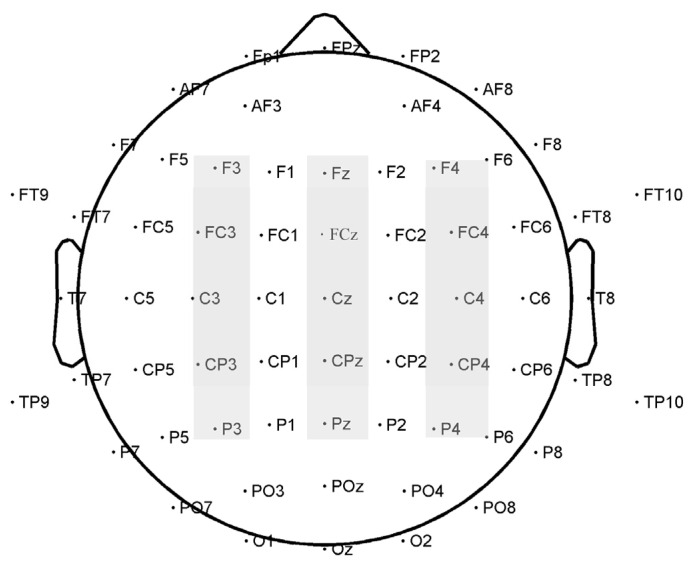
Distribution of the 64 electrodes over the scalp used to record EEG signals. The electrodes analyzed are marked in gray.

**Figure 3 brainsci-13-00855-f003:**
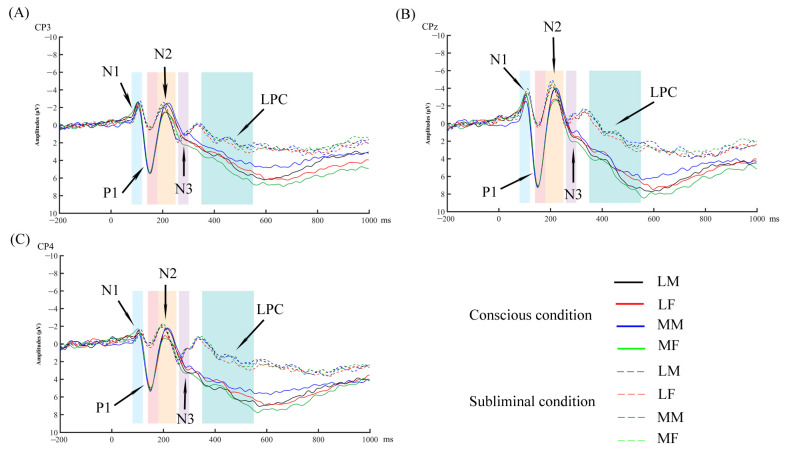
The grand average event-related potentials (ERPs) for each condition. LM represents less attractive male faces; LF represents less attractive female faces; MM represents more attractive male faces; MF represents more attractive female faces. (**A**) Grand average ERPs induced by four types of faces in conscious and subliminal conditions at electrode CP3. (**B**) Grand average ERPs induced by four types of faces in conscious and subliminal conditions at electrode CPz. (**C**) Grand average ERPs induced by four types of faces in conscious and subliminal conditions at electrode CP4.

**Figure 4 brainsci-13-00855-f004:**
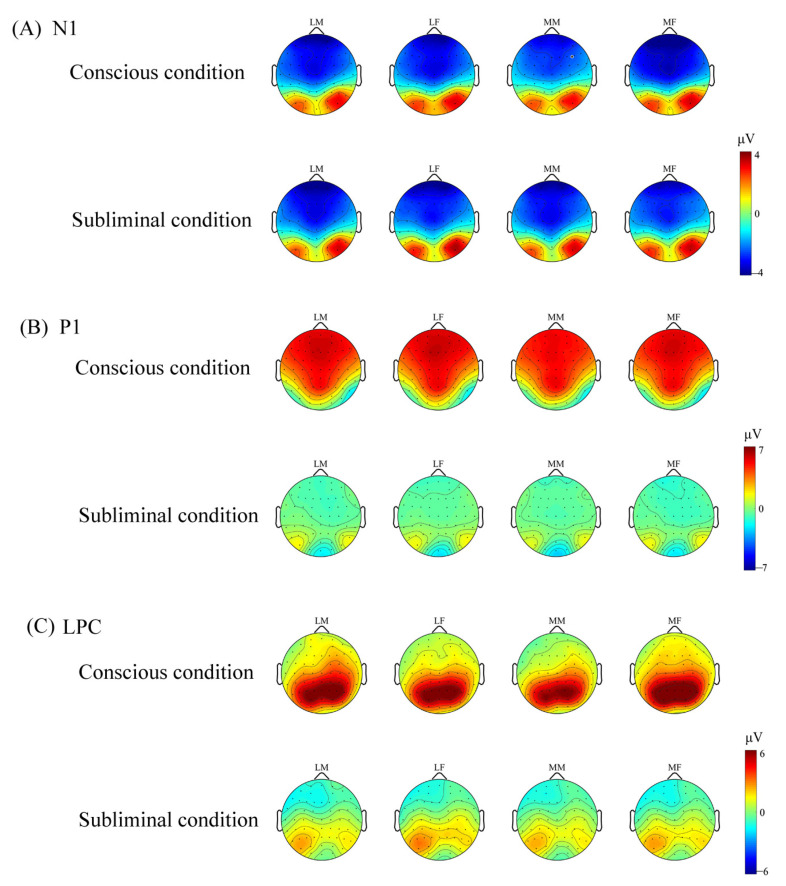
The topographic distribution of N1, P1, and LPC for each condition. LM represents less attractive male faces; LF represents less attractive female faces; MM represents more attractive male faces; MF represents more attractive female faces. (**A**) Topography of the scalp distribution of N1 component for four types of faces in conscious and subliminal conditions. (**B**) Topography of the scalp distribution of P1 component for four types of faces in conscious and subliminal conditions. (**C**) Topography of the scalp distribution of LPC component for four types of faces in conscious and subliminal conditions.

**Table 1 brainsci-13-00855-t001:** The mean (standard deviation) of the ratings of arousal, pleasure, and attractiveness of 60 experimental faces.

Stimulus Category	Arousal	Pleasure	Attractiveness
More attractive	3.67 (0.49)	4.24 (0.16)	3.31 (0.25)
Less attractive	3.54 (0.32)	4.14 (0.21)	2.78 (0.25)

**Table 2 brainsci-13-00855-t002:** The mean (standard deviation) of the ratings of arousal, pleasure, and attractiveness of practice faces.

Stimulus Category	Conscious Condition			Subliminal Condition		
	Arousal	Pleasure	Attractiveness	Arousal	Pleasure	Attractiveness
More attractive	4.47 (0.22)	5.12 (0.30)	3.81 (0.23)	4.04 (0.30)	4.72 (0.20)	3.53 (0.23)
Less attractive	3.50 (0.18)	3.36 (0.25)	2.60 (0.19)	3.40 (0.47)	3.48 (0.27)	2.74 (0.24)

## Data Availability

Data can be made available upon request to the corresponding author.

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
