# Peer review of "Neural Mechanisms of the Conscious and Subliminal Processing of Facial Attractiveness"

_brainsci, 2023, doi:10.3390/brainsci13060855_

Round 1

Reviewer 1 Report

Please see attachment. Thank you.

Please see attachment. Thank you.

Author Response

Thank you so much for your comments. We tried our best to revise the manuscript and used an English editing service (American Journal Experts; https://www.aje.cn) to correct language errors. For the problem of method, we agree with your suggestion and take it as a limitation. We hope the revisions are satisfactory. Please see the attached file for detail.

Reviewer 2 Report

The article is interesting as it easily addresses taking action on the neural mechanisms of facial attractiveness.

The experiment is well planned, since it first develops questionnaires for arousal, and attractiveness, and then uses physiological measures of brain mechanisms.   The statistics are correct, and the appropriate conclusions.

The only recommendation for the authors would be the writing, that is, it is expensive to read, and see everything that has been done. It should be structured in a more didactic way, and make it easier for the reader to read.

Thanks

Author Response

The Comments of Reviewer 2

The article is interesting as it easily addresses taking action on the neural mechanisms of facial attractiveness.

The experiment is well planned, since it first develops questionnaires for arousal, and attractiveness, and then uses physiological measures of brain mechanisms.   The statistics are correct, and the appropriate conclusions.

The only recommendation for the authors would be the writing, that is, it is expensive to read, and see everything that has been done. It should be structured in a more didactic way, and make it easier for the reader to read.

Thanks

Response:

We are grateful to you for the thoughtful comments and recommendations regarding the manuscript. In addressing them, we have revised the entire manuscript with an eye toward removing language errors and structured the manuscript in a more didactic way. We hope it is easier for the reader to read now.

Reviewer 3 Report

Abstract

-Please include the number of participants in this section.

Methods

- Please refer to the Keil et al. (2014) guidelines in Psychophysiology to ensure all required information for ERP analyses is reported.

-Provide a figure illustrating the electrodes analyzed.

Results

-I suggest including the grand average maps for each group (lower attractive, higher attractive, and conscious and subliminal). This would make it easier to understand the results.

Author Response

The Comments of Reviewer 3

Abstract

-Please include the number of participants in this section.

Response:

Thank you very much for your suggestion. We have included the number of participants in the Abstract section, please see the abstract section on Page 1.

“The 41 participants judged the facial attractiveness in a conscious condition and a subliminal condition (backward masking paradigm).”

Methods

- Please refer to the Keil et al. (2014) guidelines in Psychophysiology to ensure all required information for ERP analyses is reported.

-Provide a figure illustrating the electrodes analyzed.

Response:

Thank you very much for your suggestion. According to Keil et al. (2014), we revised the ERP recording and analysis section to ensure all required information for ERP analyses is reported. Please see Paragraph 3 on Page 6, and please also see Paragraph 2 on Page 7. We hope these revisions are satisfactory.

“All required information for ERP analyses is reported according to Keil et al. (2014). EEG signals were recorded (bandpass 0.01-30 Hz, sampling rate 500 Hz) through a 64-electrode scalp cap using the 10-20 system (Brain Products, Germany). The left and right mastoids were used as the reference. The AFz channel was used as the ground electrode. An electrode was used to measure the electrooculogram (EOG) data. Impedance was kept below 10 kΩ. The averaging of ERPs was computed off-line. EEG data were preprocessed by using Analyzer 2.0. Data were re-referenced offline to the mastoids TP9 and TP10 electrodes. Independent component analysis was applied to correct artifacts of blink. The ERPs induced by face stimuli were analyzed in the conscious and subliminal conditions. The epochs consisted of 200 ms before and 1000 ms after the onset of the face stimuli. ERPs were aligned to a 200 ms baseline. The epochs contaminated by eye blinks or muscle potentials exceeding ± 80 μV were excluded from averaging. 

Based on previous studies (Chen et al., 2012; Ma et al., 2015; Marzi & Viggiano, 2010; van Hooff, Crawford, & van Vugt, 2011; Zhang et al., 2011) and waveforms in the present study, 15 electrodes were selected for analysis: F3, Fz, and F4 (frontal sites); FC3, FCz, and FC4 (frontal-central sites); C3, Cz, and C4 (central sites); CP3, CPz, and CP4 (central-parietal sites); and P3, Pz, and P4 (parietal sites). The following components and time windows for the final analysis were chosen: N1 (80-120 ms), P1 (140-180 ms), N2 (180-250 ms), N3 (260-300 ms), and LPC (350-550 ms). Figure 2 illustrates the electrodes analyzed. Two (attractiveness: more attractive and less attractive) × 2 (face sex: male and female) × 3 (laterality: left sites, midline sites and right sites) × 5 (location: frontal, frontal-central, central, central-parietal and parietal) repeated-measures ANOVAs were conducted for the conscious and subliminal conditions. The dependent variables were the peak amplitudes and peak latencies of N1, P1 and N2 and the average amplitudes of N3 and LPC. Greenhouse‒Geisser corrections were applied where needed.”

Moreover, we provided a figure illustrating the electrodes analyzed, please see the Figure 2, in which the electrodes analyzed are marked in gray.

Results

-I suggest including the grand average maps for each group (lower attractive, higher attractive, and conscious and subliminal). This would make it easier to understand the results.

Response:

Thank you very much for your suggestion. Figure 3 shows the grand average maps for each group (lower attractive, higher attractive, and conscious and subliminal). Moreover, Figure 4 shows the topographic distribution for each condition. It is to be noted that we changed the labels of each group in the figure captions after English editing as follows: “lower attractive” was changed to “less attractive”, and “higher attractive” was changed to “more attractive”. We hope the new figures are satisfactory.

Round 2

Reviewer 1 Report

Congratulations for your hard work.

M.

Congrastulations for your hard work.

M.